# Role of Probiotics in Non-alcoholic Fatty Liver Disease: Does Gut Microbiota Matter?

**DOI:** 10.3390/nu11112837

**Published:** 2019-11-19

**Authors:** Chencheng Xie, Dina Halegoua-DeMarzio

**Affiliations:** Department of Medicine, Division of Gastroenterology and Hepatology, Thomas Jefferson University, Philadelphia, PA 19107, USA; chencheng.xie@jefferson.edu

**Keywords:** Non-alcoholic fatty liver disease, nonalcoholic steatohepatitis, probiotics, microbiome

## Abstract

Non-alcoholic fatty liver disease (NAFLD) is the hepatic consequence of metabolic syndrome, which often also includes obesity, diabetes, and dyslipidemia. The connection between gut microbiota (GM) and NAFLD has attracted significant attention in recent years. Data has shown that GM affects hepatic lipid metabolism and influences the balance between pro/anti-inflammatory effectors in the liver. Although studies reveal the association between GM dysbiosis and NAFLD, decoding the mechanisms of gut dysbiosis resulting in NAFLD remains challenging. The potential pathophysiology that links GM dysbiosis to NAFLD can be summarized as: (1) disrupting the balance between energy harvest and expenditure, (2) promoting hepatic inflammation (impairing intestinal integrity, facilitating endotoxemia, and initiating inflammatory cascades with cytokines releasing), and (3) altered biochemistry metabolism and GM-related metabolites (i.e., bile acid, short-chain fatty acids, aromatic amino acid derivatives, branched-chain amino acids, choline, ethanol). Due to the hypothesis that probiotics/synbiotics could normalize GM and reverse dysbiosis, there have been efforts to investigate the therapeutic effect of probiotics/synbiotics in patients with NAFLD. Recent randomized clinical trials suggest that probiotics/synbiotics could improve transaminases, hepatic steatosis, and reduce hepatic inflammation. Despite these promising results, future studies are necessary to understand the full role GM plays in NAFLD development and progression. Additionally, further data is needed to unravel probiotics/synbiotics efficacy, safety, and sustainability as a novel pharmacologic approaches to NAFLD.

## 1. Introduction

Non-alcoholic fatty liver disease (NAFLD) is defined by an excessive accumulation of fat in the liver tissue, when other causes of secondary hepatic fat accumulation disorders, including significant alcohol consumption, hereditary disorders, and steatogenic medication, have been ruled out [1]. NAFLD is characterized by a large spectrum of liver disease, from isolated steatosis to steatohepatitis, and can progress to cause hepatic fibrosis formation and cirrhosis. Among the NAFLD spectrum, nonalcoholic steatohepatitis (NASH) is the more severe form and presents with hepatic steatosis, lobular inflammation, and cell injury (i.e., hepatocyte ballooning) on histology [1]. NAFLD has been recognized as one of the leading causes of chronic liver disease in the world, and its prevalence, along with a global epidemic of metabolic syndrome, obesity, and insulin resistance, has risen sharply across continents [2]. A recent systemic review conducted by Younossi et al. estimated that the prevalence of NAFLD has reached 25% globally [3]. Patients with NASH are at risk of progressing to cirrhosis of the liver and/or developing hepatocellular carcinoma (HCC). The 5-year cumulative incidence of HCC, in the context of NASH fibrosis/cirrhosis, varies from 4% to 12.3% [4,5,6,7], and from 2002 to 2017 the proportion of NASH-related HCC increased 8.5-fold, rising from 2.1% to 17.9% [8].

Despite the rising prevalence of NAFLD, no definitive pharmacological treatment has been approved by the Food and Drug Administration or the European Medicines Agency. Lifestyle changes include physical activity, weight loss, and diet modification are currently the main strategies to mitigate the NAFLD epidemic. Therefore, it remains essential, though challenging, to identify new therapeutic strategies for NAFLD.

It is widely accepted that the onset and progression of NAFLD are linked to inflammation, oxidative stress, insulin resistance, dyslipidemia, and obesity [9,10]. This process is driven by the interaction of environmental, diet, and genetic factors via biochemical and immunological alterations in lipid and glucose metabolism [11]. Despite decades of research on the pathogenesis of NAFLD, the pathophysiology remains unclear and most likely with multiple mechanisms. One potential mechanism is the direct connection via the portal vein between the intestines and liver (gut-gut microbiota-liver axis). It is through this process that microbiota colonized in the intestine can modulate metabolic processes that directly or indirectly impact metabolic syndrome and its associated co-morbidities [12].

The gut microbiota (GM) has been estimated to have more than 1000 bacterial species and 100 trillion bacteria with 150-fold more genes than the human genome that colonizes in the human intestinal tract [13,14]. The two major phyla in the human intestine are *Firmicutes* (*Lactobacillus*, *Peptoniphilus*, *Ruminococcus*, *Clostridium*, and *Eubacteria*) and *Bacteroidetes* (*Bacteroides*, *Prevotella*). Other less abundant phyla include *Actinobacteria* (*Bifidobacterium*), *Proteobacteria*, and *Verrumicrobia* [15]. However, the composition and abundance of GM varies due to considerable heterogeneity between individuals and underlying conditions such as age, gender, diet, pregnancy, hormonal changes, travel, infection, and medication such as antibiotics and proton pump inhibitors [16,17,18,19]. Dysbiosis is defined as an imbalance between healthy and disease-promoting microorganisms; itis manifested through changes of diversity and fluctuation in the relative abundance of certain microorganisms [20]. The balance and homeostasis of GM is critical for maintaining health and protecting against diseases in the host.

There are a growing number of studies revealing the association between GM dysbiosis and metabolic syndrome, obesity, type 2 diabetes, NAFLD [21,22,23,24]. Zhu et al. utilized 16S ribosomal RNA sequencing and concluded that NASH patients have a distinct composition and ratio of *Firmicutes, Bacteroidetes*, and *Actinobacteria* at the level of phylum, family, and genus compared with the healthy group. NASH patients also had an increased abundance of alcohol-producing bacteria which could increase serum alcohol levels and oxidative stress, resulting in liver injury [23]. Gut microbiome changes have also been found among pediatric NAFLD subjects. Del Chierico et al. compared NAFLD, NASH, and obese pediatric patients with healthy controls and found that NAFLD patients had an increased abundance of *Anaerococcus* (*Actinobacteria*), *Ruminococcus* (*Firmicutes*), *Peptoniphilus* (*Firmicutes*), *Dorea* (*Firmicutes*), *Bradyrhizobium* (*Proteobacteria*), and *Propionibacterium acnes* (*Actinobacteria*) but a reduced abundance of *Rikenellaceae* (*Bacteroidetes*) and *Oscillospira* (*Firmicutes*). There are no significant differences in terms of microbiome composition among the NAFLD, NASH, and obese groups [25]. Furthermore, Boursier et al. identified that GM can be used as predictors for NAFLD severity and progression. The abundance of *Bacteroides* was higher in both NASH and fibrosis patients, and *Ruminococcus* abundance was higher in fibrosis patients. Through multivariate analysis Boursier et al. concluded that *Bacteroides* is independently associated with NASH and *Ruminococcus* is linked to fibrosis [26].

Despite these studies revealing an association between GM dysbiosis and NAFLD, whether gut dysbiosis is a causative factor that results in NAFLD remains unclear. Thus, further clarification is necessary to investigate the causative relationship and potential pathogenesis links between NAFLD and dysbiosis.

## 2. Pathogenesis: The Links between NAFLD and Microbiome

The pathophysiology of NAFLD/NASH progression is complicated, and the “multiple hit” hypothesis formulates the idea that multiple insults work together to facilitate the progression of disease. Some of these insults include insulin resistance, genetic and epigenetic factors, nutritional supplement, and gut microbiota [27]. Among the numerous pathophysiology mechanisms that result in NAFLD/NASH, the GM has been considered to be one of the vital contributors and has received increasing attention in recent years [28]. The liver receives portal vein circulation and is exposed to nutritional supply as well as GM derived metabolites from the gut system. The gut-liver axis characterized by a functional, bidirectional interaction between the gastrointestinal tract and liver owing to the close anatomy. The GM and liver have very complicated interactive relationships and mediated by a complex metabolic and immunologic network [29]. The primary mechanism can be summarized as changing energy harvest mode, inflammatory cytokines and related signaling pathways, altered biochemistry metabolism and GM-related metabolites (bile acid, short-chain fatty acids, aromatic amino acid derivatives, branched-chain amino acids, and ethanol; see Figure 1).

### 2.1. Interplay with Intestinal Microbiota and the Host Immune System

Many factors such as obesity, diet, alcohol intake infection, and medication, can affect the microbiome and result in impaired intestinal integrity, intestinal bacterial overgrowth, bacterial translocation, and lipopolysaccharide (LPS) releasing. The subsequent endotoxemia enters the liver through the portal circulation which induces inflammatory cytokines releasing, resulting in liver injury that may contribute to NAFLD [30]. Miele et al. indicated that NAFLD patients have a reduced expression of ZO-1 (zona occludens 1), a major tight junction protein, which results in higher intestinal permeability [31]. Volynets et al. also confirmed that intestinal permeability and endotoxin levels in plasma were significantly higher in NAFLD patients [32]. Similar results were also found in the pediatric population when Giorgio et al. reported that the severity of intestinal permeability correlates with the severity of NAFLD [33]. NAFLD patients also presented with higher rates of small intestinal bacterial overgrowth (SIBO) which correlated with the severity of steatosis [31]. Impaired integrity of the gut barrier, bacterial overgrowth, and bacterial translocation could also allow bacterial endotoxin to pass to the liver. Lipopolysaccharides (LPS) stimulate Kupffer cells via activating toll-like receptors (TLRs). These TLRs are pattern recognition receptors that recognize pathogen-associated molecular patterns (PAMPs) and damage-associated molecular patterns (DAMPs) and are inactive in healthy liver cells. Once the intestinal barrier is disrupted and endotoxemia circulates via the liver-gut through the gut-liver axis resulting in increased levels of PAMPs and DAMPs that bind with TLRs (TLR2, TLR4, TLR5, TLR9, etc.), the inflammatory cascade of releasing inflammatory cytokines (TNF-α, IL-8, IL1β) could be initiated, and lipid accumulation and cell death in hepatocytes could be stimulated leading to NAFLD, NASH, and cirrhosis [34,35,36,37,38]. Dysbiosis also plays a critical role in the weakening of mucosal immunity in a NAFLD host. Jiang et al. showed that, in the context of dysbiosis, CD4- and CD8-positive cell numbers are decreased in the duodenal mucosa lamina propria and levels of TNF-α, IL-6, and IFN-γ are increased in the NAFLD group [39].

### 2.2. Crosstalk between Intestinal Microbiota and Metabolite

#### 2.2.1. Bile Acids

Bile acid homeostasis is regulated by GM. The microbiome is involved in the synthesis of primary bile acids, cholic acid and chenodeoxycholic acid, by regulating the expression of bile acid synthesis enzymes. GM also affects other bile acid metabolism processes, includes conjugation in the liver, reabsorption in the terminal ileum, deconjugation in the small intestine, conversion to secondary bile acids (lithocholic acid and deoxycholic acid) in the colon, and enterohepatic circulation, by affecting related enzymes or transporters expression or activity [20]. The disruption of any step in bile acid metabolism could have pathophysiologic impact. Altered bile acid metabolism results in metabolic and immune effects that contribute to NAFLD [20,40,41,42]. Bile acids can regulate metabolism and inflammation through the farnesoid X receptor (FXR) and transmembrane G protein-coupled receptor 5 (TGR5). FXR is predominantly activated by primary bile acids whereas TGR5 is predominantly activated by secondary bile acids [43,44].

In terms of metabolic effect, FXR inhibits de novo lipogenesis, increases fatty acid oxidation, regulates the gene expression involved in triglyceride homeostasis that reduces steatosis and downregulates gluconeogenesis. FXR also upregulates hepatic glycogen synthesis and regulates the expression of glucose transporter 4 (GLUT-4) and glucagon-like peptide 1 (GLP-1) which affects the insulin sensitivity that is closely linked to NAFLD [43]. TGR5 also affects glucose hemostasis via inducing GLP-1 secretion which increases energy expenditure and attenuates diet-induced obesity [45].

In terms of immunologic effect, bile acids help maintain intestinal barrier integrity to protect against the GM-related inflammatory cascades in the liver [43]. Bile acids can also decrease hepatic inflammation and fibrosis via FXR and TGR5 signaling pathways [46]. A few bile acid analogs have functioned as FXR or TGR5 agonists (INT-767, WAY-362450) showing that they can reduce pro-inflammatory gene expression and increase monocyte and IL-10 production to improve hepatic inflammation, steatosis, and fibrosis [47,48].

#### 2.2.2. Short-Chain Fatty Acids (SCFAs)

SCFAs are primarily composed of acetate, propionate, and butyrate, which are microbial fermentation products that are found in the colon when a human lacks the enzymatic capacity to digest certain foods (mainly complex carbohydrates and small portions of amino acids). Most SCFAs are absorbed in the colon; however, 5–10% of SCFAs are not utilized and are excreted in the feces [20,49]. Like bile acids, SCFAs also have metabolic and immunologic effects related to NAFLD.

From a metabolic perspective, SCFAs reduce hepatic cholesterol and fatty acid synthesis while increasing hepatic lipid oxidation [50,51,52] SCFAs predominantly act on the G protein-coupled receptors GPR-41 and GPR-43, which are widely distributed in the gut enteroendocrine L cells, white adipose tissue, skeletal muscles, and the liver [20]. These L cells release glucagon-like peptides that directly act on hepatocytes by activating genes involved in fatty acid β-oxidation and insulin sensitivity which are both closely linked to NAFLD [53]. In contrast to long-chain fatty acids (LCFAs) that need carnitine shuttle to reach the mitochondrial interior, SCFAs could permeate the inner mitochondrial membrane in carnitine-independent mode. Besides, SCFAs exert weaker protonophoric effects in mitochondria and cause less disruption to the electron transport in the respiratory chain compared with LCFAs. These unique features make SCFAs important in lipogenesis, fatty acids oxidation, and ROS genesis [54].

From an immune perspective, SCFAs have an immune regulatory effect and play a complicated role in Treg cell differentiation via inhibiting the histone deacetylase and GPR-43 pathway. SCFAs have anti-inflammatory effects by reducing migration and proliferation of immune cells (T cells, neutrophil, macrophage, monocyte cells), reducing many types of pro-inflammatory cytokines (tumor necrosis factor-alpha, monocyte chemotactic protein-1, etc), and up-regulating anti-inflammatory cytokine prostaglandin E2 [55,56,57,58].

#### 2.2.3. Aromatic Amino Acid Derivatives and Branched-Chain Amino Acids

A novel class of bacterial metabolites derived from aromatic amino acids (AAA) includes tryptophan, phenylalanine, and tyrosine have recently received attention and have been considered as factors in development of NAFLD [59,60].

Tryptophan-derived bacterial metabolites are composed of indole, indole-3-propionic acid, indole-3-acetic acid, indole-3-aldehyde, tryptamine, and 3-methylindole. Among them, indole is the main component [61]. These compounds maintain intestinal integrity, reduce bacterial translocation, prevent the release of microbiota-derived components, and limit inflammatory cascades [59,60]. Oral administration of indole can reduce the expression of key genes and proteins in the LPS-induced pro-inflammatory signaling pathway, and it has also been shown to prevent LPS-induced alterations of the cholesterol metabolism in mouse models [62]. Indole-3-acetate is also found to have similar protective hepatic effects that attenuate inflammatory responses and reduce lipid accumulation in hepatocytes via the aryl hydrocarbon receptor (AhR) pathway [63].

Phenylalanine-derived bacterial metabolites are composed of phenylacetic acid, phenylpropionic acid, and benzoic acid [64]. Hoyles et al. found that plasma phenylacetic acid (PAA) levels are positively associated with steatosis severity in a morbidly obese women cohort (*n* = 56) and mice treated with PAA for 2 weeks had significantly increased triglyceride accumulation. PAA was also associated with altered gene expression involved in lipid and glucose metabolism [65].

Branched-chain amino acids (BCAA) including valine, leucine, and isoleucine, are another class of metabolites that received attention in a recent hepatic steatosis study. There is crosstalk about the relationship between microbiome, host gene expression and BCAA hepatic metabolism [65]. The BCAA biosynthesis upregulated in the context of obesity and insulin resistance [66,67]. There is also a positive correlation between hepatic steatosis and plasma and urine BCAA levels. PAA can also significantly increase hepatic BCAA utilization which can synergetically facilitate hepatic lipid accumulation [65].

#### 2.2.4. Choline

Choline metabolism is disrupted in the context of dysbiosis and deficiency in choline has been associated with NAFLD. Dysbiosis has the ability to metabolize choline to trimethylamine (TMA) which is further oxidized by hepatic flavin containing monooxygenases and converted to trimethylamine-N-oxide (TMAO) [68]. This process has two effects: (1) choline levels are reduced; (2) TMAO levels are increased.

Choline plays critical role in the process of very-low-density lipoprotein VLDL synthesis and facilitates hepatic lipid exportation. Thus choline deficiency results in triglyceride accumulation in hepatocytes [69]. A small human case study indicated that long-term total parenteral nutrition (TPN) without choline supplementation induced hepatic steatosis. After receiving parenteral nutrition solutions containing choline chloride for 6 weeks, the hepatic steatosis of subjects (4/4) resolved ultimately [70]. TMAO also increases insulin resistance, promotes inflammation and oxidative stress [71,72]. A clinical study revealed that the higher serum levels of TMAO correlated with greater severity of NAFLD [68].

#### 2.2.5. Microbial Synthesis of Ethanol

NAFLD patients have been noted to have a higher level of serum ethanol, even with the absence of alcohol consumption. Nair et al. indicated the presence of ethanol in the breath of obese females with NASH without alcohol ingestion [73]. Zhu et al. indicated that NASH patients exhibited significantly elevated blood ethanol levels compared with their lean or obese non-NASH counterparts, along with an increased abundance of ethanol-producing bacteria [23]. Similarly, Volynets et al. showed that NAFLD patients have an increased formation of endogenous ethanol in the context of dysbiosis [74].

The accumulation of endogenous alcohol from an increased abundance of alcohol-producing bacteria subsequently results in significant amounts of free radicals and reactive oxygen that cause mitochondrial dysfunction, hepatic cellular inflammation and injury [23,75,76]. Other mechanisms that ethanol plays a role in and that contribute to NAFLD development include, downregulating tight junction expression, up-regulating de novo lipogenesis, decreasing fatty acid oxidation, and reducing very low density lipoprotein (VLDL) exportation from the liver [20].

### 2.3. Energy Extraction and Consumption Balance Disrupted by GM

GM is thought to alter the energy influx and expenditure which may contribute to NAFLD development. Turnbaugh et al. conducted a study with a cohort of 154 obese and lean twins and found that in obese humans, the GM-enriched phosphotransferase system is involved in the microbial processing of carbohydrates, which yields an enhanced capacity to extract calories from the diet [77]. Additionally, the microbiome also altered the host’s energy expenditure via metabolic products such as bile acids and SCFAs. As mentioned earlier, dysbiosis can alter the level of bile acids and SCFAs and ultimately affect energy extraction and expenditure. It has been found that bile acids can increase whole-body energy expenditure and energy consumption via the activation of brown adipose tissue [78]. SCFAs themselves can provide an energy source for different cell types and tissues. Butyrate can be utilized by colonocytes as an energy substrate, whereas acetate and propionate can be utilized as substrates in the process of glucose and fatty acid synthesis, respectively [79]. Canfora et al. infused SCFA mixtures (either acetate, butyrate, or propionate) into the colon of overweight/obese men. The SCFAs mixture increased fat oxidation, energy expenditure, and attenuated fasting free glycerol concentrations compared with the placebo group [80].

## 3. Clinical Application of Probiotics and Synbiotics in NAFLD/NASH Patients

Given the correlation between dysbiosis and liver damage, the growing burden of NAFLD, and lacking effective pharmacologic interventions, restructuring GM to reverse dysbiosis seems like a potential therapeutic strategy. Probiotics are non-pathogenic, live microorganisms that provide a health benefit to the host by modifying GM when delivered in sufficient quantities [81]. The most frequently mentioned probiotics in recent clinical trials are *Lactobacilli*, *Streptococci*, and *Bifidobacteria*. Prebiotics are non-digestible carbohydrates that can be fermented by bacteria and subsequently change GM composition and activity to promote health benefits [82]. Synbiotics refer to the combination of prebiotics and probiotics.

Probiotics and synbiotics can normalize GM and reverse dysbiosis, which could potentially benefit NAFLD patients [83]. Multiple experimental trials have shown probiotics to have significant therapeutic effects in fatty liver mice models. The trials showed that administrating probiotics to high-fat-diet fed mice could prevent the onset of hepatic steatosis and improve hepatic steatosis and fibrosis. The protective effects have multiple mechanisms including reducing hepatic lipid deposition, decreasing endotoxemia, reducing oxidative stress, anti-inflammatory effects through the modulation of nuclear factor kappa B (NF-κB) and tumor necrosis factor (TNF), and antifibrotic effects through the modification of transforming growth factor-beta (TGF-β) and collagen expression [84,85,86,87,88]. These preclinical studies suggest that probiotics therapy could be a potential pharmacologic intervention for NAFLD patients, though the animal model have their limitations. For example, mice are germ-free animals and the intestine bacterial composite and abundance are different from that of a human.

Most human clinical trials conducted to study the therapeutic effect of probiotics/synbiotics in NAFLD patients have been conducted over the past 10 years and are small scale with mixed clinical outcomes. However, the overall outcome indicates that probiotics/synbiotics could be a promising therapeutic strategy for the NAFLD population. Table 1 summarizes 26 randomized controlled trials (RCT) that compared probiotics and/or synbiotics versus placebo in NAFLD patients from January 2010to June 2019 in [69,89,90,91,92,93,94,95,96,97,98,99,100,101,102,103,104,105,106,107,108,109,110,111,112,113].

### 3.1. Biochemistry Evaluation

Most of the clinical trials in Table 1 include serum aspartate aminotransferase (AST) and alanine aminotransferase (ALT) level changes to assess the effects of probiotics or synbiotics on liver function. A double-blind RCT enrolled 30 patients with NAFLD, thehe intervention group received 3 months of a combination of *Lactobacillus bulgaricus* and *Streptococcus thermophilus* (500 million per day) and had significant AST and ALT improvement [89]. Similarly, in a 72 NAFLD patients double-blinded RCT, Nabvi et al. utilized probiotic yogurt (300 g/day) containing *Lactobacillus acidophilus La5* and *Bifidobacterium lactis Bb12* as an intervention for 8 weeks. They concluded that probiotics could significantly improve AST and ALT in NAFLD patients when compared with the placebo group [96]. Shavakhi et al. [92] used probiotics capsules (*Lactobacillus acidophilus*, *Lactobacillus casei*, *Lactobacillus rhamnosus*, *Lactobacillus bulgaricus*, *Bifidobacterium breve*, *Bifidobacterium longum*, and *Streptococcus thermophilus*) plus metformin 1000mg per day as an intervention.

When compared with the control group who only take metformin 1000mg per day, it was found that the probiotics with metformin synergically improved liver aminotransferases in NASH patients. A few other RCTs that utilized different species of probiotics also revealed that probiotics could improve either AST or ALT, or both transaminases in NAFLD patients [69,91,93,97,101,105,110,112]. Four recent synbiotics RCTs utilized a combination of probiotics (*Lactobacillus acidophilus*, *Lactobacillus casei*, *Lactobacillus rhamnosus*, *Lactobacillus bulgaricus*, *Bifidobacterium breve*, *Bifidobacterium longum*, and *Streptococcus thermophilus*) plus prebiotics (either fructooligosaccharide or inulin) and concluded that synbiotics supplements can also benefit liver function [95,103,106,111].

Similar to the adult population, pediatric clinical trials also revealed that probiotics and synbiotics could improve AST and ALT in NAFLD patients. Vajro et al. enrolled 20 pediatric obese NAFLD patients and offered the intervention group a *Lactobacillus rhamnosus* strain GG (12 billion CFU/day). The data indicated a significant decrease in ALT after an 8 weeks trial [90]. Similarly, in a triple blinded RCT, Famouri et al. enrolled 64 obese children with sonographic NAFLD and the interventional group received probiotics capsules (*Lactobacillus acidophilus*, *Bifidobacterium lactis*, *Bifidobacterium bifidum*, and *Lactobacillus rhamnosus*). The primary outcome indicated that the probiotics group had a significantly lower mean value of AST and ALT, as well as lower levels of cholesterol, triglycerides, and LDL-C after a 12 week intervention [104]. Miccheli et al. enrolled 31 pediatric NAFLD patients with a VSL#3 (*Streptococcus thermophilus*, *Bifidobacterium breve*, *Bifidobacterium longum*, *Bifidobacterium infantis*, *Lactobacillus acidophilus*, *Lactobacillus plantarum*, *Lactobacillus casei*, and *Lactobacillus bulgaricus*) intervention for 4 months. They concluded that, compared with the placebo group, AST but not ALT significantly improved [97].

However, not all of the completed trials support the notion that probiotics or synbiotics can improve liver enzyme levels. For example, Ahn et al. published a trial in 2019 that enrolled 68 adult obese NAFLD patients. The interventional group was treated with a multispecies probiotics mixture (*Lactobacillus acidophilus*, *Lactobacillus rhamnosus*, *Lactobacillus paracasei*, *Pediococcus pentosaceus*, *Bifidobacterium lactis*, and *Bifidobacterium breve*) for 12 weeks. Results revealed that probiotics improved the intrahepatic fat (IHF) fraction measured by MRI-PDFF but there was no significant improvement of ALT or AST when compared with the placebo group [113]. Alisi et al. conducted a RCT in 2014, enrolling 22 NAFLD children. The interventional group was treated with probiotics capsules (*Streptococcus thermophilus*, *Bifidobacterium breve*, *Bifidobacterium longum*, *Bifidobacterium infantis*, *Lactobacillus acidophilus*, *Lactobacillus plantarum*, *Lactobacillus casei*, and *Lactobacillus bulgaricus*) for 4 months. However, the trial failed to reveal any significant liver enzyme improvement when compared with the placebo group [94]. Some synbiotics trials have also failed to show liver enzyme improvement. For instance, Ferolla et al. conducted an RCT of 50 NASH patients and indicated that synbiotics (*Lactobacillus reuteri* + inulin) neither improved liver function nor steatosis [99].

These mixed results also occurred with other serum markers including Gamma-glutamyl transferase (GGT), alkaline phosphatase (ALP), total cholesterol (TC), triglyceride (TG), low-density lipoprotein cholesterol (LDL-C), high-density lipoprotein cholesterol (HDL-C), fasting serum glucose (FBS) and inflammatory markers [69,89,90,91,92,93,94,95,96,97,98,99,100,101,102,103,104,105,106,107,108,109,110,111,112,113]. Given the pattern of mixed results, Table 2 summarizes the most recently published meta-analyses of RCTs with probiotic/synbiotic treatment for NAFLD/NASH populations [114,115,116,117,118].

All of the meta-analyses listed indicated that microbial therapies are associated with a significant improvement in ALT and AST [114,115,116,117,118]. Regarding the other serum markers, Liu et al. included 15 probiotics and synbiotics RCTs involving 782 patients with NAFLD up to April 2018. They found that probiotic and synbiotic supplementation is associated with a significant reduction in TG, TC, HDL-C, LDL-C, and TNF-α levels [117]. Koutnikova et al. included 111 RCTs representing 6826 subjects with metabolic syndrome, type II DM, and NAFLD. Subgroup analysis of NAFLD patients did not indicate that probiotics significantly reduced GGT [116]. Khan et al. included 12 probiotics and synbiotics RCTs up to June 2018. They concluded that a reduction in CRP occurred in synbiotics trials. However, TNF-α, LDL-C, TG, and TC only significantly improved in the synbiotics but not probiotics group, and HDL-C and FBS did not significantly change in either group [115]. Of note, serologic markers have high variability and not directly reflect the disease progression in context of NAFLD. Thus, there are emerging studies utilized the imaging modalities to evaluate NAFLD clinical outcome outline below.

### 3.2. Imaging Modalities to Assess Hepatic Steatosis and Stiffness

Several trials included imaging evaluation as part of clinical outcomes to assess the effect of probiotics and synbiotics on hepatic steatosis, fibrosis, and liver stiffness. Ultrasonography (US) is the most commonly used imaging modality for liver morphology study. The research teams of Shavakhi [92], Miccheli [97], Asgharian [98], Famouri [104], and Bakhshimoghaddam et al. [107] reported that US-measured steatosis significantly improved in adult patients who took probiotics and synbiotics supplements. Alisi et al. revealed a similar efficacy of probiotics improving US-measured hepatic steatosis in pediatric NAFLD patients [94]. Manzhalii et al. conducted an RCT with 75 enrolled NASH patients on low-fat/low-calorie diets. Probiotics (*Lactobacilli*, *Bifidobacteria*, and *Streptococcus thermophilus*) were administered in the intervention group for 12 weeks. US-measured liver stiffness improved in the probiotics group compared with the control group [105]. However, in 2018, Wang et al. got different results in a double-blind RCT with 200 NAFLD patients. The experimental group received *Bifidobacterium*, *Lactobacillus*, *Enterococcus*, and *Bacillus* subtilis for a 1-month duration and there was no significant difference in the severity of steatosis measured by US [112]. The likely explanation was that the intervention duration was relatively shorter. Also, as a measurement tool, US has technical limitations when it comes to accurately evaluating the severity of hepatic steatosis.

Over recent years, elastography-based imaging techniques have received increased attention in the field of non-invasive assessment for organ tissue, especially the liver. These techniques utilize soft tissue elasticity changes in various pathologies to yield qualitative and quantitative analyses and are superior to US in assessing the severity of liver steatosis, fibrosis, and cirrhosis [119]. Eslamparast et al. conducted a double-blind RCT that enrolled 52 NAFLD patients with a synbiotics (probiotics cocktail - *Lactobacillus casei*, *Lactobacillus rhamnosus*, *Lactobacillus acidophilus*, *Lactobacillus bulgaricus*, *Bifidobacterium breve*, *Bifidobacterium longum*, *Streptococcus thermophilus*, and *the prebiotic fructooligosaccharide*) intervention for 28 weeks and the elastography-measured fibrosis score improved [95]. Mofidi et al. applied the same intervention as the Eslamparasts team, enrolling 50 NAFLD patients who took part in a transient elastography evaluation. Results indicated that synbiotics are associated with hepatic steatosis and fibrosis improvement which is consistent with the results of Eslamparasts team [106]. Kobyliak conducted two double-blind RCTs in 2018. One enrolled 58 NAFLD with type 2 diabetes patients. The intervention team received probiotics called “Symbiter” which is composed of *Bifidobacterium*, *Lactobacillus*, *Lactococcus*, *Propionibacterium*, and *Acetobacter* for 8 weeks and utilized shear wave elastography (SWE) to measure hepatic fibrosis. The trial concluded that probiotics can improve fatty liver index but that there is no significant change in liver stiffness graded by SWE [110]. His team conducted another double-blind RCT that enrolled 48 NAFLD with type 2 diabetes patients. The intervention team received a combination of the same probiotics mentioned above plus Omega-3 fatty acids called “Symbiter Omega”. After an 8-week intervention, results revealed similar significantly reduce liver steatosis but not liver stiffness [109].

Magnetic resonance imaging–derived proton density fat fraction (MRI-PDFF) is another novel modality that, over recent years, has been used to evaluate hepatic steatosis. The feature of accurately measuring the distribution of intrahepatic fat fraction (IHF) across the liver in seconds has led to its rising popularity in NAFLD research [120]. A few trials utilized this technique to analyze the efficacy of probiotics/synbiotics in steatosis, for example, Ahn et al. concluded that a probiotics therapy group had a reduced mean value IHF compared with the control group, however, there was no significant change in liver stiffness between the probiotics and control groups [113]. Ferolla et al. reported that MRI-PDFF-measured steatosis was reduced in the synbiotics group but there was no significant change in liver fibrosis [99]. There is only one published RCT that utilized proton-magnetic resonance spectroscopy to measure intrahepatic triglyceride content (IHTG). Wong et al. conducted a trial with 20 NASH patients. The intervention group was treated with *Lactobacillus plantarum*, *Lactobacillus deslbrueckii*, *Lactobacillus acidophilus*, *Lactobacillus rhamnosus*, and *Bifidobacterium bifidum* for 6 months. The primary outcome of the RCT is that IHTG decreased from the baseline of 22.6% to 14.9% in the probiotics group whereas IHTG remained static in the usual care group [93].

Two large-scale meta-analyses conducted by Sharpton et al. [118] and Liu et al. [117] indicated that probiotics/synbiotics improved liver stiffness that was measured by elastography, and probiotics/synbiotics are associated with hepatic steatosis improvement as graded by the US though analyses showed heterogeneity. Another meta-analysis conducted by Khan et al. indicated that probiotics/synbiotics are associated with a significant improvement in liver fibrosis score graded by fibroscan [115].

### 3.3. Histologic Evaluation

The fact that most probiotics/synbiotics intervention trials lack a liver biopsy to further assess histological changes in liver steatosis, fibrosis, cirrhosis, and inflammation is a noticeable limitation. Nevertheless, there are still two RCTs that utilized liver biopsy at the end of clinical trials to evaluate histological changes after probiotics and synbiotics interventions. Malaguarnera et al. conducted an RCT that enrolled 66 NASH patients. The study used synbiotics (*Bifidobacterium longum*+fructooligosaccharides) as an intervention for 24 weeks and the liver biopsy indicated hepatic steatosis improvement and reduced NASH activity in the synbiotics group when compared with the control group [91]. Duseja et al. conducted a double-blind RCT with 30 enrolled NAFLD patients. The probiotics administered were composed of *Lactobacillus paracasei*, *Lactobacillus plantarum*, *Lactobacillus acidophilus*, *Lactobacillus delbrueckii subsp*. *bulgaricus*, *Bifidobacterium longum*, *Bifidobacterium infantis*, *Bifidobacterium breve*, and *Streptococcus thermophilus*. After 1 year of treatment, a liver biopsy indicated significantly reduced hepatocyte ballooning, NAS score, and fibrosis compared with the control group. However, there was no significant hepatic steatosis or lobular inflammation difference between the two groups [69].

### 3.4. Safety and Tolerability of Probiotics in NAFLD Patients

Probiotics/synbiotics have a relatively safe profile, and historically, probiotics have been widely used in dairy products without significant safety concerns. The Agency for Healthcare Research and Quality (AHRQ) released a report in 2011 that extensively reviewed literature regarding the probiotic safety and adverse event reports. It included six probiotic genera (*Lactobacillus*, *Bifidobacterium*, *Streptococcus*, *Enterococcus*, *Bacillus*, and *Saccharomyces*) and 622 studies and concluded that the existing probiotic RCTs reveal no evidence of increased risk. However, due to the lack of assessment and systematic reporting of adverse events in probiotic intervention studies, AHRQ was not equipped to clarify the safety of probiotic interventions with confidence [121]. Though clinical trials rarely reported adverse effects or issues of safety with patients with NAFLD, lacking long-term, post-intervention follow-up in clinical trials is a setback regarding safety assessment. There are a few case reports of probiotics having adverse effects such as systemic infections (bacteremia and endocarditis), bowel ischemia, inflammatory reaction, D-lactic acidosis, and gastrointestinal side effects [122]. Although microbial therapy seems to be safe, it is necessary to evaluate probiotic- or synbiotic-associated adverse effects and investigate safety profiles for long-term use.

## 4. Concluding Remarks

With the rapidly growing incidence and prevalence of NAFLD and lack of effective pharmacological intervention, there is an urgent need for developing novel medications for treatment. The link between GM and NAFLD has received increased attention and a growing number of studies have revealed that the host-gut microbiome interaction and pathophysiology of dysbiosis may promote the development of NAFLD. Bacterial derivative metabolites, immune-related effects, and energy extraction/consumption balance disruption might promote NAFLD progression. Therefore, restoring dysbiosis could be a potential therapeutic strategy for NAFLD, an idea backed by multiple recent probiotics and synbiotics clinical trials in the specific context of NAFLD and NASH. The literature to date is encouraging and supports the notion that probiotics/synbiotics may improve transaminase levels, hepatic steatosis, and NAFLD activity score. To some extent, probiotics/synbiotics can also reduce proinflammatory cytokines such as TNF-α and the interleukin family (IL-1, IL-6, IL-8).

Overall, probiotics/synbiotics have a safe profile and are well-tolerated though more safety evaluations need to be completed in the future. Additionally, the sustainability of the probiotic/synbiotic protective effect on NAFLD over the long-term is still not clear. In this growing NAFLD epidemic era, further studies are necessary to more clearly elucidate the efficacy, safety, and sustainability of probiotics/synbiotics in the management of NAFLD.

## Figures and Tables

**Figure 1 nutrients-11-02837-f001:**
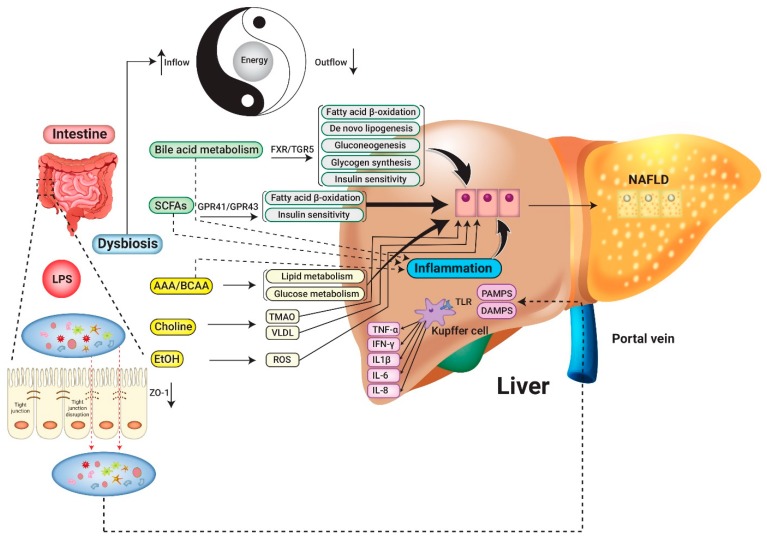
The roles gut microbiota play in liver steatosis. LPS: lipopolysaccharide; SCFAs: short-chain fatty acids; AAA: aromatic amino acids; BCAA: branched-chain amino acids; EtOH: ethanol; FXR: farnesoid X receptor; TGR5: transmembrane G protein-coupled receptor 5; GPR: G protein-coupled receptor; TMAO: trimethylamine-N-oxide; VLDL-C: very-low-density lipoprotein cholesterol; ROS: Reactive oxygen species; TNF-α: tumor necrosis factor-alfa; IFN- γ: Interferon-gamma; IL-1β: interleukin 1beta; IL-6: interleukin 6; IL-8: interleukin 8; TLR: toll-like receptor; NAFLD: Non-alcoholic fatty liver disease.

**Table 1 nutrients-11-02837-t001:** Summary of major randomized controlled trials using probiotics or synbiotics therapy for NAFLD/NASH.

Study	Design/Population (*N* = Included in the Trial)	Bacterial Species	Duration	Main Outcome Related to NAFLD	
				Serology	Imaging or Biopsy
Aller et al. [89], 2011	Double-blind RCT/NAFLD (*N* = 28)	*Lactobacillus bulgaricus* *Streptococcus thermophilus*	3 months	(↓) ALT, AST, GGT (-) Anthropometric parameters and cardiovascular risk factors	N/A
Vajro et al. [90], 2011	Double-blind RCT/NAFLD obese children (*N* = 20)	*Lactobacillus rhamnosus*	8 weeks	(↓) ALT (-) BMI, TNF-α, peptidoglycanpolysaccharide antibody	N/A
Malaguarnera et al. [91], 2012	RCT/NASH (*N* = 66)	*Bifidobacterium longum* *+* Prebiotics (Fructo-oligosaccharides)	24 weeks	(↓) AST, LDL-C, CRP, TNF-α, endotoxin, HOMA-IR (-) ALT, bilirubin, HDL-C, TC, TG, glucose, insulin, C-peptide, BMI	The liver biopsy indicated that steatosis and NASH activity improved
Shavakhi et al. [92], 2013	Double-blind RCT/NASH on metformin (*N* = 64)	*Lactobacillus acidophilus* *Lactobacillus casei* *Lactobacillus rhamnosus* *Lactobacillus bulgaricus* *Bifidobacterium breve* *Bifidobacterium longum* *Streptococcus thermophilus*	6 months	(↓) ALT, AST, TG, TC, BMI (-) FBS	Grading of steatosis based on US measurement improved
Wong et al. [93], 2013	RCT/NASH (*N* = 20)	*Lactobacillus plantarum, Lactobacillus deslbrueckii* *Lactobacillus acidophilus Lactobacillus rhamnosus Bifidobacterium bifidum*	6 months	(↓) ALT (-) AST, BMI, waist circumference, glucose, and lipid levels	Proton-magnetic resonance spectroscopy-measured that intrahepatic triglyceride content (IHTG) improved
Alisi et al. [94], 2014	Double-blind RCT/NAFJD children (*N* = 44)	*Streptococcus thermophilus* *Bifidobacterium breve* *Bifidobacterium longum Bifidobacterium infantis Lactobacillus acidophilus Lactobacillus plantarum Lactobacillus casei Lactobacillus bulgaricus*	4 months	(↓) BMI (↑) GLP-1, activated GLP-1 (-) ALT, triglycerides, HOMA	Grading of steatosis based on US measurement improved
Eslamparast et al. [95], 2014	Double-blind RCT/NAFLD with lifestyle modification (*N* = 52)	*Lactobacillus casei* *Lactobacillus rhamnosus* *Lactobacillus acidophilus* *Lactobacillus bulgaricus* *Bifidobacterium breve* *Bifidobacterium longum* *Streptococcus thermophilus* *+* prebiotic (fructo-oligosaccharide)	28 weeks	(↓) ALT, AST, CRP, TNF-ɑ, NF-κB p65 (-) BMI	Transient elastography- measured fibrosis score improved
Nabvi et al. [96], 2014	Double-blind RCT/NAFLD (*N* = 72)	*Lactobacillus acidophilus* *Bifidobacterium lactis*	8 weeks	(↓) ALT, AST, TC, LDL-C (-) Glucose, TG, HDL-C	N/A
Miccheli et al. [97], 2015	Triple-blind RCT/NAFLD children (*N* = 31)	*Streptococcus thermophilus* *Bifidobacterium breve* *Bifidobacterium longum Bifidobacterium infantis Lactobacillus acidophilus Lactobacillus plantarum Lactobacillus casei Lactobacillus bulgaricus*	4 months	(↓) AST, total and active GLP-1, BMI (-) ALT, TG, TC, HDL-C, LDL-C, glucose, insulin	Grading of steatosis based on US measurement improved
Asgharian et al. [98], 2016	Double-blind RCT/NAFLD (*N* = 80)	*Lactobacillus casei* *Lactobacillus acidophilus* *Lactobacillus rhamnosus* *Lactobacillus bulgaricus* *Bifidobacterium breve* *Bifidobacterium longum* *Streptococcus thermophilus* *+* prebiotic (fructo-oligosaccharide)	8 weeks	Prevent ASL and ALT elevation (-) CRP, BMI	Grading of steatosis based on US measurement improved
Ferolla et al. [99], 2016	RCT/NASH (*N* = 50)	*Lactobacillus reuteri* *+* prebiotic (inulin)	3 months	(↓) BMI (-) AST, ALT, ALP, GGT, TC, TG, HDL-C, VDL-C, LPS, and intestinal permeability measured by lactulose/mannitol urinary excretion	MRI-PDFF- measured steatosis improved but liver fibrosis had no significant change
Sepideh et al. [100], 2016	Double-blind RCT/NAFLD (N = 42)	*Lactobacillus casei* *Lactobacillus acidophilus* *Lactobacillus rhamnosus* *Lactobacillus bulgaricus* *Bifidobacterium breve* *Bifidobacterium longum* *Streptococcus thermophilus*	8 weeks	(↓) IL-6, FBS, insulin, insulin resistance (-) TNF-alpha	N/A
Abdel et al. [101], 2017	RCT/NASH with obesity (N = 30)	*Lactobacillus acidophilus*	1 month	(↓) ALT, AST (-) TG, TC, FBS Bilirubin, HDL-C	N/A
Behrouz et al. [102], 2017	Double-blind RCT/NAFLD (N = 89)	*Lactobacillus casei* *Lactobacillus rhamnosus* *Lactobacillus acidophilus* *Bifidobacterium longum* *Bifidobacterium breve*	12 weeks	(↓) Leptin, insulin, and HOMA-IR (-) Adiponectin, FBS	N/A
Ekhlasi et al. [103], 2017	Double-blind RCT/NAFLD with Vitamin E (N = 60)	*Lactobacillus casei* *Lactobacillus rhamnosus Stretococcus thermophilus* *Bifidobacterium breve Lactobacillus acidophilus* *Bifidobacterium longum Lactobacillus bulgaricus* *+* prebiotic (fructo-oligosaccharide)	8 weeks	(↓) ALT, AST, ALP, sytolic BP, malondialdehyde, TNF-alpha (-) Diastolic Blood Pressure, nitric oxide, BMI	N/A
Famouri et al. [104], 2017	Triple-Blind RCT/NAFLD obese children (N = 64)	*Lactobacillus acidophilus* *Bifidobacterium lactis* *Bifidobacterium bifidum* *Lactobacillus rhamnosus*	12 weeks	(↓) ALT, AST, cholesterol, triglycerides, LDL-C, and waist circumference (-) BMI, weight	Grading of steatosis based on US measurement improved
Manzhalii et al. [105], 2017	RCT/NASH with low-fat/low-calorie diet (N = 75)	*Lactobacilli* *Bifidobacteria* *Streptococcus thermophilus*	12 weeks	(↓) ALT, BMI and cholesterol (-) GGT	Liver stiffness based on US measurement improved
Mofidi et al. [106], 2017	Double-blind RCT/NAFLD (N = 50)	*Lactobacillus acidophilus* *Lactobacillus casei* *Lactobacillus rhamnosus* *Lactobacillus bulgaricus* *Bifidobacterium breve* *Bifidobacterium longum* *Streptococcus thermophilus* *+* prebiotic (fructo-oligosaccharide)	28 weeks	(↓) AST, ALT, GGT, glucose, triglyceride, Total cholesterol, CRP, NF-κB p65 (-) HDL-C, LDL-C, TNF-α	Transient elastography-measured hepatic steatosis and fibrosis improved
Bakhshimoghaddam et al. [107], 2018	RCT/NAFLD (N = 102)	*Bifidobacterium animalis* *+* prebiotic (inulin)	24 weeks	(↓) AST, ALT, GGT, ALP, TG, TC	Grading of steatosis based on US measurement improved
Javadi et al. [108], 2018	Double-blind RCT/NAFLD (N = 75)	*Bifidobacterium longum* *Lactobacillus acidophilus* *+* prebiotic (inulin)	3 months	(↓) CRP, TNF-α, BMI (↑) TAC (-) IL-6, MDA	N/A
Kobyliak et al. [109], 2018	Double-blind RCT/NAFLD with type II DM (N = 48)	*Bifidobacterium* *Lactobacillus* *Lactococcus* *Propionibacterium* *Acetobacter* *+* omega-3 fatty acids	8 weeks	(↓) GGT, TG, TC, VLDL-C, TNF-α, IL-6 (-) AST, ALT, LDL-C, HDL-C, INF-γ, IL-1β, IL-8	Shear Wave Elastography-measured fatty liver index improvement but no significant change in liver stiffness
Kobyliak et al. [110], 2018	Double-blind RCT/NAFLD with type II DM (N = 58)	*Bifidobacterium* *Lactobacillus* *Lactococcus* *Propionibacterium* *Acetobacter*	8 weeks	(↓) AST, GTT, TNF-α, IL-6 (-) ALT, TC, TG, VLDL-C, HDL-C, IL-1β, IL-8, IFN- γ	Shear Wave Elastography-measured fatty liver index improvement but no significant change in liver stiffness
Sayari et al. [111], 2018	RCT/NAFLD with sitagliptin (N = 138)	*Lactobacillus casei Lactobacillus rhamnosus* *Lactobacillus acidophilus* *Lactobacillus bulgaricus Bifidobacterium breve Bifidobacterium longum* *Streptococcus thermophilus* *+* prebiotic (fructo-oligosaccharide)	16 weeks	(↓) AST, TC, LDL-C, FBS (-) ALT, HDL-C, TG, BMI	N/A
Wang et al. [112], 2018	Double-blind RCT/NAFLD (N = 200)	*Bifidobacterium Lactobacillus* *Enterococcus* *Bacillus subtilis*	1 month	(↓) AST, ALT, TC, TG, glucose, VLDL-C, TNF-α (↑) High molecular weight adiponectin (-) HDL-C	Grading of steatosis based on US measurement showed no significant difference
Ahn et al. [113], 2019	Double-blind RCT/NAFLD with obesity (N = 68)	*Lactobacillus acidophilus* *Lactobacillus rhamnosus* *Lactobacillus paracasei* *Pediococcus pentosaceus Bifidobacterium lactis* *Bifidobacterium breve*	12 weeks	(↓) TG (-) AST, ALT, TC, HDL-C, glucose, insulin, TNF-α, IL-6, LPS	MRI-PDFF-measured intrahepatic fat fraction (IHF) and mean IHF reduced compared with control, but no significant change in liver stiffness
Duseja et al. [69], 2019	Double-blind RCT/NAFLD (*N* = 30)	*Lactobacillus paracasei* *Lactobacillus plantarum* *Lactobacillus acidophilus* *Lactobacillus delbrueckii* *subsp. bulgaricus* *Bifidobacterium longum* *Bifidobacterium infantis* *Bifidobacterium breve* *Streptococcus thermophilus*	1 year	(↓) ALT, ALP, leptin, TNF-α, and LPS (-) AST, Bilirubin, Adiponectin, IL-1β, IL-6	The biopsy indicated hepatocyte ballooning, NAS score and fibrosis improved, but not steatosis or lobular inflammation

NAFLD: Non-alcoholic fatty liver disease; NASH: Nonalcoholic steatohepatitis; ALT: alanine transaminase; AST: aspartate aminotransferase; GGT: gamma-glutamyl transferase; ALP: alkaline phosphatase; LDL-C: low-density lipoprotein cholesterol; HDL-C: high-density lipoprotein cholesterol; TC: total cholesterol; TG: triglyceride; FBS: fasting serum glucose; GLP-1: glucagon-like peptide 1; BMI: body mass index; LPS: Lipopolysaccharides; CRP: C reactive protein; TNF-α: tumor necrosis factor-alfa; NF-κB p65: nuclear factor kappa-light-chain-enhancer of activated B cells subunit p65; IFN- γ: Interferon-gamma; IL-1β: interlukin 1beta; IL-6: interleukin 6; IL-8: interleukin 8; MDA: malondialdehyde; TAC: total antioxidant capacity; NAS: NAFLD activity score; HOMA-IR: homeostasis model assessment of insulin resistance; US: ultrasonography; MRI-PDFF: magnetic resonance imaging–derived proton density fat fraction; IHF: intrahepatic fat fraction.

**Table 2 nutrients-11-02837-t002:** Summary of recent meta-analysis of RCTs with probiotics/synbiotics therapy for NAFLD/NASH patients.

Study	Population	Study Period	Conclusions
Loman et al. [114], 2018	Included 25 studies (most are RCT): Among them, 9 studies used prebiotics, 11 studies used probiotics, and 7 studies used synbiotics. 1309 patients were included.	Up to December 14, 2017	Microbial therapies significantly reduced AST and ALT, but not CRP. The results of serum cholesterol and LDL-C are mixed among prebiotics, probiotics, and synbiotics.
Khan et al. [115], 2019	Included 12 probiotics/synbiotics RCTs for NAFLD. 748 patients were included.	Up to June 10, 2018	Probiotics/synbiotics were associated with a significant improvement in ALT, AST, and liver fibrosis score graded by fibroscan. There was a reduction in CRP with synbiotics. The TNF-α, LDL-C, TG, and TC significantly improved with synbiotics but not with probiotics in a subgroup analysis. There were no significant changes in HDL-C, HOMA-IR, or FBS in either group.
Koutnikova et al. [116], 2019	Included 105 articles with 111 RCTs representing 6826 subjects (includes metabolic syndrome, type II DM, and NAFLD patients). Among them, about 658 patients had NAFLD.	January 1990 to June 2018	In subjects with fatty liver diseases, probiotics reduced AST and ALT, but not GGT.
Liu et al. [117], 2019	Included 15 probiotics and synbiotic RCTs, involving 782 patients with NAFLD.	Up to April 2018	Probiotics and synbiotics supplementation could significantly improve AST, ALT, TG, TC, HDL-C, LDL-C, homeostasis model assessment-insulin resistance, TNF-α, liver steatosis, and liver stiffness. However, probiotics and synbiotics could not ameliorate BMI, waist circumference, or FBS.
Sharpton et al. [118], 2019	Included 21 RCTs (1252 participants) with NAFLD. 9 trials evaluated probiotics and 12 trials evaluated synbiotics.	January 1, 2005 to December 1, 2018	Probiotics/synbiotics could improve AST and ALT. Probiotics/synbiotics were also associated with hepatic steatosis improvement when graded with ultrasound. Last, probiotics/synbiotics were associated with liver stiffness improvement when measured by elastography, although analyses showed heterogeneity. Probiotics, but not synbiotics, were associated with a significant reduction in body mass index.

NAFLD: Non-alcoholic fatty liver disease;DM: diabetes mellitus; ALT: alanine transaminase; AST: aspartate aminotransferase; GGT: gamma-glutamyl transferase; LDL-C: low-density lipoprotein cholesterol; HDL-C: high-density lipoprotein cholesterol; TC: total cholesterol; TG: triglyceride; FBS: fasting serum glucose; BMI: body mass index; CRP: C reactive protein; TNF-α: tumor necrosis factor-alfa; HOMA-IR: homeostasis model assessment of insulin resistance.

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
