# Peer review of "Role of Probiotics in Non-alcoholic Fatty Liver Disease: Does Gut Microbiota Matter?"

_nutrients, 2019, doi:10.3390/nu11112837_

Round 1
Reviewer 1 Report
The manuscript is very interesting as it reviews the actual knowledge about the association between gut microflora (GM) dysbiosis and non-alcoholic fatty liver disease (NAFLD) with an in-depth description of the pathological mechanisms and clinical trials that are currently available to explain this crosstalk.
The article is clearly laid out.
The Word "in" is missing in the title: Role of probiotics in nonalcoholic....
The abstract clearly describes the manuscript content and findings in the field.
Results are written in a correct sequence and figures and tables are correct and consistent with the article content. In Table 1, a new column should be added to include the clinical relevance of the otucomes which could allow a rapid evaluation of the serology and imaging results.
The conclusión paragraph is written in a reasonable way and is related to previous findings and future perspective.
English language is correct, although the punctuation sign "coma" should be deleted before the Word "and" in several sentences.
Cited references are updated and accurate.
All my comments express my best opinión about the manuscript quality. Only a few corrections have been marked in the text.

Author Response
Re: Requested Article Revision- Role of Probiotics in Non-alcoholic Fatty Liver Disease: does gut microbiota matter?
Dear Nutrients Editor and Reviewers,
We appreciate the careful review and constructive suggestions. Following this letter are the reviewers' comments with point-to-point responses. We also refined the language and revisions made to the manuscript are marked using track changes.
We want to extend our appreciation for taking the time and effort to provide such insightful guidance. Please let us know if any further clarification is needed. Thank you for your time and consideration.
Sincerely,
Chencheng Xie, MD
Dina Halegoua-DeMarzio, MD
Point-by-point response:
Reviewer #1,
The word "in" is missing in the title: Role of probiotics in nonalcoholic....
Response: Thank you for the suggestions. As suggested, we added the missing “in” to the title.
In Table 1, a new column should be added to include the clinical relevance of the outcomes which could allow a rapid evaluation of the serology and imaging results.
Response: We appreciate the reviewer’s suggestions. However, it is difficult to evaluate the outcome clinical relevance of these pilot studies due to the small number of study subjects. If the reviewer agrees, we would like to maintain the current table format. If the reviewer insists that the additional column to assess clinical relevance of outcome is essential, would the reviewer mind showing us a few examples or elaborating a little more details regarding the methodology, and we would be happy to adjust the table in second round of revision.
English language is correct, although the punctuation sign "coma" should be deleted before the Word "and" in several sentences.
Response: We appreciate the reviewer’s comments. We have made the appropriate changes.
Reviewer 2 Report
The review article by Xie, Halegoua-DeMarzio provides a quick overview of the impact of microbiota (GM) the pathogenesis of non-alcoholic fatty liver disease (NAFLD) and non-alcoholic steatohepatitis (NASH). The review summarizes relevant signaling cascades perturbed in NAFLD/NASH, significant metabolites produced by GM, diagnostic biomarkers, and liver imaging modalities. The tables are helpful and summarize the most relevant clinical studies. The authors should explain the multiple-hit hypothesis of NAFLD/NASH and the enterohepatic cycle better. Overall the content is sound, and the figure summarizes major standard pathways involved in the pathogenesis of NAFLD/ NASH. The English language, however, needs major revision.
Specific comments:
Why is there an “and between the last author’s name and the academic title? The title “… Do bacteria matter” should be more specific and include “ …Do gut microbiota matter?” What impact has the enterohepatic cycle on GM? Are the changes in the GM a cause for NASH or a consequence of it? Please discuss this aspect in more detail. Please use consistent formatting for non-alcoholic fatty liver disease (NAFLD). Please mention the multiple-hit hypothesis of NAFLD/NASH and explain it. The mitochondrial metabolism of short-chain fatty acids also differs from long-chain fatty acids. Please briefly mention this aspect.
Author Response
Re: Requested Article Revision- Role of Probiotics in Non-alcoholic Fatty Liver Disease: does gut microbiota matter?
Dear Nurtients Editor and Reviewers,
We appreciate the careful review and constructive suggestions. Following this letter are the reviewers' comments with point-to-point responses. We also refined the language and revisions made to the manuscript are marked using track changes.
We want to extend our appreciation for taking the time and effort to provide such insightful guidance. Please let us know if any further clarification is needed. Thank you for your time and consideration.
Sincerely,
Chencheng Xie, MD
Dina Halegoua-DeMarzio, MD
Point-by-point response:
Reviewer #2,
1. The English language, however, needs major revision.
Response: We appreciate the reviewer’s comments. As suggested, Dr. Halegoua-DeMarzio, who is a native English speaker, re-read the article again and proofread fixed the language issues in the main text.
2. Why is there an “and” between the last author’s name and the academic title?
Response: We appreciate the reviewer’s comments. It was a typo, and we corrected it in the revision.
3. The title “… Do bacteria matter” should be more specific and include “ …Do gut microbiota matter?”
Response: Thank you for the suggestions. As suggested, we changed the title.
4. What impact has the enterohepatic cycle on GM? Are the changes in the GM a cause for NASH or a consequence of it? Please discuss this aspect in more detail.
Response: We appreciate the reviewer’s comments. We mentioned it in our manuscript in the last paragraph in the section of “1. Introduction”-“Despite these studies revealing an association between GM dysbiosis and NAFLD, whether gut dysbiosis is a causative factor that results in NAFLD remains unclear. Thus, further clarification is necessary to investigate the causative relationship and potential pathogenesis links between NAFLD and dysbiosis.” We reviewed the currently available literature, and it lacks convincing evidence to verify the causative relationship between NAFLD and dysbiosis, and further studies are needed for full elucidation. We also added a few sentences in the first paragraph at the section of “2. Pathogenesis: The links between NAFLD and microbiome” to address the interaction of GM and the gut-liver axis.
5. Please use consistent formatting for non-alcoholic fatty liver disease (NAFLD).
Response: We appreciate the reviewer’s comments. We have revised the content in the text according to the reviewer’s suggestions.
6. Please mention the multiple-hit hypothesis of NAFLD/NASH and explain it.
Response: We appreciate the reviewer’s comments. We have added the content in the text according to the reviewer’s suggestions. Please see the first paragraph in “2. Pathogenesis: The link between NAFLD and the microbiome”.
7. The mitochondrial metabolism of short-chain fatty acids also differs from long-chain fatty acids. Please briefly mention this aspect.
Response: We appreciate the reviewer’s comments. We have added the content in the text according to the reviewer’s suggestions. Please see the second paragraph in “2.2.2 Short-chain fatty acids (SCFAs)”.
Reviewer 3 Report
This is a well written and comprehensive review and I have no major comments.
Minor points/typos
Typo line 4 - misplaced "and"
Typo line 135 - should be "the microbiome is" etc
Section 2.2.1 bile acids - It is mentioned that certain microbes will metabolise host bile acids to secondary BA, it might be worth exploring this further as increased abundance of 7a dehydroxylating bacteria has been demonstrated in conditions "adjacent" to NAFLD such as obesity (https://www.tandfonline.com/doi/full/10.1080/19490976.2017.1406584 & others) and could simultaneously lead to increased cholesterol uptake and increased intestinal permeability due to these toxic intermediates (e.g. https://www.ncbi.nlm.nih.gov/pmc/articles/PMC5449568/)
Section 3.1 Biochemistry - is it worth highlighting that ALT/AST are extremely variable measures so may be of limited use in conditions such as NAFLD - there does seem to be a good number of studies mentioned using TE or ultrasound which I would regard as much more useful endpoints.
Author Response
Re: Requested Article Revision- Role of Probiotics in Non-alcoholic Fatty Liver Disease: does gut microbiota matter?
Dear Nutrients Editor and Reviewers,
We appreciate the careful review and constructive suggestions. Following this letter are the reviewers' comments with point-to-point responses. We also refined the language and revisions made to the manuscript are marked using track changes.
We want to extend our appreciation for taking the time and effort to provide such insightful guidance. Please let us know if any further clarification is needed. Thank you for your time and consideration.
Sincerely,
Chencheng Xie, MD
Dina Halegoua-DeMarzio, MD
Point-by-point response:
Reviewer #3
1. Typo line 4 - misplaced "and"
Response: Thank you for the suggestions. As suggested, we deleted the “and”.
2. Typo line 135 - should be "the microbiome is" etc
Response: Thank you for the suggestions. As suggested, we have revised the content in the text according to the reviewer’s recommendations.
3. Section 2.2.1 bile acids - It is mentioned that certain microbes will metabolize host bile acids to secondary BA, it might be worth exploring this further as increased abundance of 7a dehydroxylating bacteria has been demonstrated in conditions "adjacent" to NAFLD such as obesity (https://www.tandfonline.com/doi/full/10.1080/19490976.2017.1406584 & others) and could simultaneously lead to increased cholesterol uptake and increased intestinal permeability due to these toxic intermediates (e.g. https://www.ncbi.nlm.nih.gov/pmc/articles/PMC5449568/)
Response: We appreciate the reviewer’s comments. We have added the content in the text according to the reviewer’s suggestions. Please see the first paragraph in “2.2.1 Bile Acid”.
4. Section 3.1 Biochemistry - is it worth highlighting that ALT/AST are extremely variable measures so may be of limited use in conditions such as NAFLD - there does seem to be a good number of studies mentioned using TE or ultrasound which I would regard as much more useful endpoints.
Response: We appreciate the reviewer’s comments. We have added the content in the text according to the reviewer’s suggestions. Please see the last paragraph in the “3.1 Biochemistry evaluation”.
Round 2
Reviewer 2 Report
The authors revised the manuscript appropriately. There are still several typos and sometimes spaces between words are missing. Please carefully double-check and correct.
Author Response
November 13, 2019
Re: Requested Article Revision- Role of Probiotics in Non-alcoholic Fatty Liver Disease: does gut microbiota matter?
Dear Nutrients Editor and Reviewers,
We appreciate the careful review and constructive suggestions. Following this letter are the reviewers' comments with point-to-point responses. We also refined the language and revisions made to the manuscript are marked using track changes.
We want to extend our appreciation for taking the time and effort to provide such insightful guidance. Please let us know if any further clarification is needed. Thank you for your time and consideration.
Sincerely,
Chencheng Xie, MD
Dina Halegoua-DeMarzio, MD
Point-by-point response:
Reviewer #2,
The authors revised the manuscript appropriately. There are still several typos and sometimes spaces between words are missing. Please carefully double-check and correct.
Thank you for your evaluation. A careful review was completed with correction of these errors.